# Hydroxycinnamic Acid Derivatives from Coffee Extracts Prevent Amyloid Transformation of Alpha-Synuclein

**DOI:** 10.3390/biomedicines10092255

**Published:** 2022-09-12

**Authors:** Maria Medvedeva, Natalia Kitsilovskaya, Yulia Stroylova, Irina Sevostyanova, Ali Akbar Saboury, Vladimir Muronetz

**Affiliations:** 1Faculty of Bioengineering and Bioinformatics, Lomonosov Moscow State University, 119234 Moscow, Russia; 2Department of Systems Biology, National Medical Research Center for Obstetrics, Gynecology and Perinatology of Ministry of Healthcare of Russian Federation, 117997 Moscow, Russia; 3Federal State Budgetary Institution “Federal Research and Clinical Center of Physical-Chemical Medicine”, Federal Medical Biological Agency, 119435 Moscow, Russia; 4Belozersky Institute of Physico Chemical Biology, Lomonosov Moscow State University, 119234 Moscow, Russia; 5Institute of Biochemistry and Biophysics, University of Tehran, Tehran 1417614335, Iran

**Keywords:** alpha-synuclein, amyloid transformation, coffee, neurodegenerative diseases

## Abstract

Earlier we showed that derivatives of hydroxycinnamic acids prevent amyloid transformation of alpha-synuclein and prion protein. The aim of this work was to determine the content of 3-hydroxycinnamic acid derivatives in coffee extracts and to evaluate their activity in relation to alpha-synuclein amyloid aggregation. Hydroxycinnamic acid derivatives were identified in aqueous and ethanol extracts of coffee beans by quantitative mass spectrometric analysis. Only 3,4-dimethoxycinnamic acid (13–53 μg/mL) was detected in significant amounts in the coffee extracts, while ferulic acid was present in trace amounts. In addition, 3-methoxy-4-acetamidoxycinnamic acid (0.4–0.8 μg/mL) was detected in the roasted coffee extracts. The half-maximum inhibitory concentrations of alpha-synuclein fibrillization reaction in the presence of coffee extracts, as well as inhibitory constants, were determined using thioflavin T assay. The inhibitory effect of black and green coffee extracts on alpha-synuclein fibrillization is dose-dependent, and in a pairwise comparison, the constants of half-maximal inhibition of fibrillization for green coffee extracts are comparable to or greater than those for black coffee. Thus, coffee extracts prevent pathological transformation of alpha-synuclein in vitro, probably due to the presence of 3,4-dimethoxycinnamic acid in them. Consequently, coffee drinks and coffee extracts can be used for the prevention of synucleinopathies including Parkinson’s disease.

## 1. Introduction

Natural aromatic compounds are known to prevent the pathological transformation of amyloidogenic proteins, which makes them promising candidates for creating drugs for neurodegenerative diseases of amyloid nature, such as Parkinson’s and Alzheimer’s diseases [1,2]. Among natural compounds, curcumin is best known for its various effects on many pathological processes [3,4], including neurodegenerative ones [5,6,7]. The main limitation of curcumin application is its insolubility in water, which makes it necessary to resort to different methods to increase its bioavailability [8]. However, among natural compounds, there is a whole class of substances whose structure is similar to curcumin, more precisely “half” of its molecule, at the same time being highly soluble in water. These compounds include a variety of natural and synthetic derivatives of cinnamic acid. For example, 3,4DMCA (3,4-dimethoxycinnamic acid) has higher solubility in water than curcumin. In our studies, it has been shown that some cinnamic acid derivatives prevent the pathological transformation of amyloidogenic proteins (alpha-synuclein and prion protein) [9,10,11,12]. 

Interactions and anti-prion activity have been studied for several cinnamic acids, including 3,4DMCA, whose effect has been considerable on the biologically common compounds [9,10]. The ability of hydroxycinnamic acids to partially inhibit the oligomerization and formation of prion amyloid fibrils has been confirmed by various physicochemical methods, such as isothermal titration calorimetry, dynamic light scattering, thioflavin T fluorescence, and circular dichroism spectroscopy [9,10].

In our works, using the methods of circular dichroism, molecular docking, thioflavin T fluorescence, proteinase K proteolysis, and ion-conducting microscopy, we also showed that three of nine studied synthetic and natural hydroxycinnamic acid derivatives effectively prevented alpha-synuclein amyloid transformation: ferulic acid (FA) ≈ 3-methoxy-4-acetamidoxycinnamic acid (3M4ACA) >> 3,4DMCA, with half-maximal inhibitory concentrations of 13 ± 2 μM, 50 ± 2 μM, and 251 ± 41 μM, respectively [11]. We also suggested that the mechanism of derivatives action may involve the formation of non-amyloid aggregates incapable of further fibrillization [11]. A number of other natural phenolic and polyphenolic substances are also capable of stabilizing the native unfolded conformation of amyloidogenic proteins. There are several possible pathways, including inhibition of amyloid protofilaments and α-synuclein fibrils formation [13,14], destabilization of oligomers and α-synuclein fibrils assembly [15], or remodeling and inactivation of toxic amyloid oligomers [13] due to the presence of aromatic and hydroxyl groups on the ligand phenyl ring. Aromatic polyphenol rings can presumably interact with monomeric and oligomeric forms of α-synuclein, sterically inhibiting further protein aggregation [16].

It should be especially noted that among the studied cinnamic acid derivatives exhibiting an anti-amyloid effect there were natural compounds, namely ferulic acid [15], caffeic acid [17], rosmarinic acid [15,18], chlorogenic acid [19,20], sinapinic acid [21], and 3,4-dimethoxycinnamic acids [9,11]. Champagne wine extracts rich in these components have been shown to protect neurons against 5-S-cysteinyl-dopamine in vitro [22]. Hydroxycinnamic acid derivatives are ubiquitous in the cell wall of plants, including food plants, where they are formed in the course of shikimate pathway reactions and account for up to one-third of phenolic derivatives that come to humans with food [23,24]. It is also important that caffeic acid, ferulic acid, 5-O-feruloylquinic acid, and 3,4DMCA are not only found in food, but are also metabolites present in human and animal tissues [25,26,27]. Moreover, coffee consumption is known to cause a significant increase in the concentration of these compounds in the blood [27,28,29,30,31]. Thus, the natural origin of cinnamic acid derivatives facilitates their use as anti-amyloid agents suitable for the prevention and treatment of neurodegenerative amyloidosis. In addition, previously reported results show that cinnamic acid derivatives could exert their anti-amyloid effect not only when used individually as purified compounds, but also as part of food products containing these compounds.

The aim of this work was to identify cinnamic acid derivatives in coffee bean extracts using quantitative mass spectrometric analysis and to check the anti-amyloid effect of coffee extracts on the pathological transformation of alpha-synuclein. In addition, the task of the work was to compare the features of the cinnamic acid derivatives’ extraction with different methods of processing coffee beans and to confirm the effectiveness of these compounds not only in an individual state, but also as a part of a complex mixture of substances.

## 2. Materials and Methods

In the present work, we used two samples of natural derivatives of cinnamic acid, 3,4DMCA and 3-methoxy-4-hydroxycinnamic acid (ferulic), as standards in our mass spectrometric experiments. These samples were obtained by our colleagues by synthesis. The identity of the natural compounds was proved by our colleagues earlier. 3,4-dimethoxycinnamic acid and thioflavin T were commercial (Sigma, Burlington, MA, USA). Solutions of cinnamic acid derivatives were prepared on PBS (pH 7.4) at a concentration of 5 mM and stored at 4 °C.

### 2.1. Preparation of Coffee Extracts

Coffee extracts were prepared from green and medium-roasted coffee beans from a local supermarket. Whole green coffee beans were ground for 45 s in a coffee grinder. A total of 3 g of ground green or black coffee was mixed with 12 mL of milliQ water or 12 mL of 40% ethanol. The samples were left to stir for 1 h with a magnet stirrer. In addition, samples of green and black coffee in milliQ water were brought to a boil on the lowest setting of the hotplate. After sample preparation, the coffee extracts were centrifuged for 10 min at rcf 12,100× *g* on an Eppendorf MiniSpin centrifuge and the pellets were discarded.

### 2.2. Purification of Recombinant Alpha-Synuclein

Recombinant alpha-synuclein (full-length wild-type protein without additional motifs with a codon substitution from TAC to TAT, also encoding tyrosine, in order to avoid mis-insertion of cysteine in the bacterial system during translation [32]) was expressed in *E. coli* BL21 (DE3) cells and purified according to a previously described procedure [33]. After acidification of the cell extract with 9% HCl (pH 2.8), the precipitated proteins were removed by centrifugation (15,000× *g*, 5 min, 4 °C). The pH value of the supernatant was corrected to 7.5 using 1 M potassium phosphate solution, pH 11. Ammonium sulfate was added to the supernatant to 40% saturation, and the suspension was left overnight at 4 °C until a precipitate formed. After the double wash with 40% ammonium sulfate solution and further centrifugation, the protein suspension was stored at 4 °C. We used an extinction coefficient A^0.1%^ 0.412 mL/(mg*cm) for measurements of protein concentration.

### 2.3. Fibrillation of Alpha-Synuclein

Synucleinopathies are characterized by the accumulation of alpha-synuclein aggregates. Fibrillation of alpha-synuclein in vitro allows us to partially simulate these conditions and test the effect of coffee extracts on the formation of fibrils. There are various protocols for obtaining alpha-synuclein fibrils; the method used in this article was used in our earlier works [11,33].

A solution of alpha-synuclein in ammonium sulfate was centrifuged, and the precipitate was dialyzed against PBS buffer, pH 4.0. The dialyzed protein solutions were diluted to a final concentration of 0.4 mg/mL (27.6 μM) with the same buffer, and 0.3 mL of each was transferred to 9 mL glass tubes. To measure IC50 values, the fibrils were grown in the presence of cinnamic acid derivatives at 37 °C and with constant stirring at 300 rpm for 48 h. After 48 h of incubation, 5 μL of an aliquot was taken for thioflavin T fluorescence analysis to monitor the formation of amyloid aggregates.

### 2.4. Thioflavin T Fluorescence Analysis

Thioflavin T (ThT) fluorescence was detected on a CLARIOstar plate reader (BMG LAB-TECH GmbH, Ortenberg, Germany), mixing 5 μL aliquots from samples with growing fibrils with 95 μL of 25 μM ThT solution in FLUOTRAC 200 black (Greiner) 96-well plates. Fluorescence intensity was measured in triplicate after 10 min incubation at 20 °C using 440 nm excitation and 490 nm emission wavelengths.

### 2.5. Determination of the Half-Maximum Inhibitory Concentration (IC50)

An amount of 0.05 mL of each coffee extract at various concentrations from 0 to 250 mg/mL was added to 0.3 mL of 0.4 mg/mL (27.6 µM) alpha-synuclein monomer solution. The alpha-synuclein fibrils were then grown for 2 days according to the procedure described earlier. The dependence of ThT fluorescence on the concentration of the coffee extract was used to calculate IC50. IC50 values were calculated by approximation with OriginPro 8.6 (OriginLab, Northampton, MA, USA) [34].

### 2.6. Determination of Inhibitory Constant (Ki) for Coffee Extracts

The inhibition constant was calculated by the formula Ki = [E] × [I]/[EI], where [E] is the concentration of the unbound inhibitor protein (3,4DMCA), [I] is the concentration of the unbound inhibitor (3,4DMCA), and [EI] is the concentration of the inhibitor-bound protein. In our case different [I] and their corresponding fluorescence levels are known (they are proportional to [E]) [35,36]. Letting [EI] be represented as [E]_0_ − [E], where [E]_0_ is the total alpha-synuclein concentration (i.e., the protein concentration in the absence of the inhibitor), then we have
Ki = [E] × [I]/([E]_0_ – [E]),
Ki × ([E]_0_ – [E]) = [E] × [I],
[E]_0_/[E] = 1 + 1/Ki × [I].

If there is no inhibitor ([I] = 0), then [E]_0_/[E] = 1. If the amount of inhibitor in the system increases, then [E]_0_/[E] also increases. We obtain a linear dependence of [E]_0_/[E] on [I] with a slope angle equal to 1/Ki.

The E_0_/E values were determined as the ratio of fluorescence levels in the absence of the inhibitor and in the presence of a certain concentration of the inhibitor. Values of concentrations of 3,4DMCA were determined knowing its concentration in a particular coffee extract according to the results of mass spectrometry. Approximations of the dependence of E_0_/E on the concentration of 3,4DMCA were built in the program OriginPro 8.6 (OriginLab) [34].

### 2.7. Mass Spectroscopic Analysis

Samples were studied by multiple reaction monitoring—this mode is based on the extraction from the mass-spectrum of the chromatographic ion fraction of the component to be determined, and its subsequent fragmentation and registration of the most intense and at the same time characteristic daughter ion. This transition from the detection of the parent ion to the detection of the daughter ion is called an MRM transition, which, in addition to the m/z values of the parent and daughter ions, is also set by the fragmentation energy.

Studies were performed on a QTRAP 5500 mass spectrometer, ABSciex (ABSciex, Vaughan, ON, Canada) in ESI + mode with an Agilent 1260 chromatography system (Agilent, Santa Clara, CA, USA). An Agilent Zorbax Eclipse Plus C18 3.0 × 50 mm 1.8-micron analytical column (Agilent, Santa Clara, CA, USA) was used for chromatography. To calibrate the instrument, 3-methoxy-4-acetamidoxycinnamic, ferulic, and 3,4-dimethoxycinnamic acids in 50 mM concentrations dissolved in methanol were used and MRM transitions were determined for them using Analyst 1.6.3 software (AB Sciex, Canada) (Table 1).

The conditions for the mass spectrometric analysis were as follows: ESI + source; voltage of 5500 V; nitrogen as the atomizing and drying gas; atomizing gas pressure of 50 conventional units; drying gas pressure of 50 conventional units; temperature of +300 °C; and scanning type-multiple reaction monitoring (MRM).

Further, for the quantitative determination of the acid content, chromatographic conditions were selected (Appendix A)—the mobile phase consisted of deionized water (A) and acetonitrile (B) (99.9%) containing 0.1% formic acid each, and calibration dependences were constructed for solutions of known concentration in the methanol/water system in a 1:1 ratio (Appendix A).

A 5 µL sample was injected for the analysis. The study was carried out at a column temperature of 30 °C. Samples with unknown concentrations were prepared by diluting the samples 10-fold with a methanol/water system in the ratio of 1:1.

Then we used MultiQuant 3.0.1 software (AB Sciex, Canada) to construct calibration dependences and calculate confidence parameters for the calibrants (solutions with known concentration), and used them to calculate the concentration of the substances under study for the samples being analyzed (see Section 3, Table 3).

## 3. Results

We determined the content of three cinnamic acid derivatives, ferulic acid, 3-methoxy-4-acetamidoxycinnamic acid, and 3,4DMCA, in coffee extracts, for which effective inhibition of alpha-synuclein amyloid transformation was shown [11]. Chemical structures of these derivatives, similar to half of the curcumin molecule, are presented in Table 2.

The content of the three cinnamic acid derivatives in the coffee extracts was studied by the MRM method using a mass spectrometer with a chromatographic system. Figure 1 shows the standard spectra of the three compounds: 3-methoxy-4-acetamidoxycinnamic acid (3M4ACA), ferulic acid (FA), and 3,4-dimethoxycinnamic acid (3,4DMCA). Knowing the “retention time” (the time taken for a compound to pass through the chromatographic system) determined for each standard, its presence in the mixture can be confirmed. For 3-methoxy-4-acetamidoxycinnamic acid, the retention time was 1.5 min; for ferulic acid, 2.5 min; and for 3,4-dimethoxycinnamic acid, 4 min.

Figure 2 shows chromatograms of an aqueous extract of black coffee. The vertical axis shows the intensity of the MRM transitions, while the horizontal axis shows the retention time of the substance from the chromatographic column (RT). Signals corresponding to 3-methoxy-4-acetamidoxycinnamic, ferulic, and 3,4-dimethoxycinnamic acids are marked; RT is recorded according to the RT of standards, respectively. Similar spectra were obtained in the analysis of the green and black coffee extracts obtained under other conditions (Appendix A). It was shown that all three studied hydroxycinnamic acid derivatives were present in all samples. Based on calibration curves for the studied hydroxycinnamic acid derivatives (Figure 3), their content in the coffee extracts was determined (Table 3).

The data in Table 3 show that all the coffee extracts contained high concentrations of 3,4-dimethoxycinnamic acid (25–53 µg/mL) and very insignificant amounts of ferulic acid (0.13–3.0 ng/mL). Interestingly, the synthesized 3-methoxy-4-acetamidoxycinnamic acid, whose presence in natural objects was unknown, was also detected in coffee extracts, from 0.02 to 0.81 µg/mL. Extraction with ethanol is not more efficient compared to aqueous extraction, and heating also does not increase the yield of the metabolites of interest. The content of 3,4DMCA in the black coffee extracts compared with the green coffee extracts was about twice as high, and 4M4ACA was 15–40 times higher. An inverse picture was observed for ferulic acid, which was significantly (6–15 times) higher in the green coffee extracts in comparison to black coffee.

On the presented spectra of coffee extracts, except for the three studied compounds, there are many additional “peaks” corresponding to other compounds in all the extracts of black coffee (Figure 1 and Figure 2). However, their content is significantly lower than the content of 3,4-dimethoxycinnamic acid.

### 3.1. Determination of the Half-Maximal Inhibition Constant of Alpha-Synuclein Fibrillization for Coffee Extracts

The half-maximum inhibition constant of the alpha-synuclein fibrillization reaction was determined in the presence of six samples of different coffee extracts (Figure 4).

The fluorescence of thioflavin T was shown to drop with an increasing concentration of coffee extracts. Thus, the inhibitory effect of black and green coffee extracts on α-synuclein fibrillization is dose-dependent. The constants of half-maximal inhibition for aqueous extracts of black and green coffee were 2.62 mg/mL and 2.34 mg/mL, respectively; for aqueous extracts obtained by boiling, they were 3.47 mg/mL and 4.68 mg/mL for black and green coffee, respectively; and for 40% alcoholic extracts, they were 1.48 mg/mL (for black coffee) and 16.3 mg/mL (for green coffee). In a pairwise comparison, the constants of half-maximal inhibition of fibrillization for the green coffee extracts were comparable to or greater than those for black coffee. Since the main trends in the efficiency of alpha-synuclein fibrillization inhibition by black and green coffee extracts are opposite to the data on the ratio of the contents of the studied metabolites, except for ferulic acid, other compounds probably show anti-amyloid activity in green coffee, whose extraction efficiency is increased by using ethanol.

### 3.2. Determination of the Inhibition Constant Ki

Ki inhibition constants of alpha-synuclein fibrillization were also determined based on the concentration of 3,4DMCA in the studied extracts. The data given in Table 4 show that the inhibition constant of alpha-synuclein fibrillization was of the order of 10^−6^ mol/L. According to literature data for curcumin, for example, this value is between 10^−5^ and 10^−7^ mol/L [36]. Differences in Ki between coffee extracts prepared by different methods are probably due to the presence of other compounds with anti-amyloid activity against alpha-synuclein. However, the main effect preventing the pathological transformation of alpha-synuclein is related to the action of 3,4DMCA.

## 4. Discussion

In our previous studies, we have shown that three of nine studied synthetic and natural cinnamic acid derivatives effectively prevented alpha-synuclein amyloid transformation: ferulic acid (FA), 3-methoxy-4-acetamidoxycinnamic acid (3M4ACA), and 3,4DMCA. We suggested that the cinnamic derivatives are involved in the formation of non-amyloid aggregates incapable of further fibrillization [11]. In this study, we were focused on the bioavailability of these three derivatives in coffee extracts and on the efficiency of the coffee extracts, obtained with different procedures (aqueous or ethanol extracts with or without boiling).

First of all, we determined the content of three cinnamic acid derivatives, ferulic acid (0.13–3.0 ng/mL), 3-methoxy-4-acetamidoxycinnamic acid (0.02 to 0.81 µg/mL), and 3,4DMCA (25–53 µg), in different coffee extracts using mass spectroscopic analysis. Of particular note is the presence of 3-methoxy-4-acetamidoxycinnamic acid in black coffee extracts, which has been previously obtained only by synthesis. It is possible that 3M4ACA is formed during the roasting of coffee beans, which allows black coffee to be used as a source of this compound. Interestingly, the content of 3,4DMCA was about twice as high in the black coffee extracts compared to the green coffee extracts, and 4M4ACA was 15–40 times higher, while the concentration of ferulic acid was higher in the green coffee extracts. For comparison, those who drink several cups of coffee per day might easily ingest 500–800 mg of cinnamic acids [29]. The highest obtained IC50 value among our extracts was the one for the ethanol extract of green coffee—16.3 mg/mL.

Determination of the half-maximal inhibition constant of alpha-synuclein fibrillization for coffee extracts and the inhibition constant Ki for 3,4DMCA from different coffee extracts was used to compare the efficiency of different coffee samples. Ki reflects the binding affinity and IC50 represents the functional strength of the inhibitor. The concentration of cinnamic acids correlates to IC50, e.g., the ethanol extract of green coffee contained the lowest concentrations of cinnamic acids and showed the largest value of IC50 (16.3 mg/mL). In a pairwise comparison, the IC50 values of fibrillization for the green coffee extracts were comparable to or greater than those for black coffee. Since there is no strict correlation between IC50 and the ratio of the studied metabolites’ contents, we can make an assumption that other compounds present in coffee possess anti-amyloid activity. According to Table 4, the IC50 for 3,4DMCA was between 0.9 and 4.5 μM. Which, although close to the values obtained for the pure compound, is still significantly lower. It is likely that other cinnamic acid derivatives and various compounds present in coffee extracts have an additional effect. This is indicated by different IC50 values for different types of extracts (Table 4). It is likely that the composition of compounds exerting anti-amyloid action in these extracts is different. The inhibition constant of alpha-synuclein fibrillization for 3,4DMCA in the different coffee extracts was of the order of 10^−6^ mol/L.

Thus, we have shown that the derivatives of cinnamic acid, mainly 3,4-dimethoxycinnamic acid, are present in coffee extracts, obtained similarly to the classic recipe for making a coffee drink. At the same time, coffee extracts prevent the pathological transformation of alpha-synuclein, probably due to the presence of 3,4-dimethoxycinnamic acid in them. Therefore, cinnamic acid derivatives can have an anti-amyloid effect not only individually, but also as a part of complex mixtures. Finding out the features of the cinnamic acid derivatives extraction allows us to propose optimal methods for obtaining medications enriched with these compounds from coffee beans, which can be used for the prevention and treatment of synucleinopathies and other amyloidoses. These experiments show that the three studied compounds, because of their high water solubility, can be extracted even in water without heating from both types of grains.

To date, a significant number of works have been conducted confirming the neuroprotective effect of coffee and some of its components (e.g., caffeic acid, i.e., 3,4-dihydroxycinnamic acid) in neurodegeneration, but it is clear that the mechanisms are quite multifaceted and require confirmation. In particular, this has been shown in cellular and mouse models of Parkinson’s disease [37,38]. In described experiments, administration of caffeic acid for 8 weeks alleviated motor deficits, induced autophagy, reduced A53T-synuclein accumulation, and reduced the loss of dopaminergic neurons in the substantia nigra of A53T transgenic mice (i.e., mice expressing the human mutant A53T-synuclein) [39]. The neuroprotective effects of coffee and some of its components have been repeatedly proved in relation to the amyloid [40,41,42]. Our study shows that one of the presumed mechanisms to reduce the accumulation of pathological alpha-synuclein aggregates, and the resulting neurotoxicity and neurodegeneration, is the direct inhibition of the fibrillization process. This is clearly indicated by the reproducibility of the previously shown inhibitory effect of 3,4-dimethoxycinnamic acid, which is a methylated metabolite of caffeic acid. We have shown that 3,4-dimethoxycinnamic acid is present in mimicking the preparation of coffee extracts, and such extracts effectively reduce the rate of alpha-synuclein fibrillization.

Champagne wine extracts are another source of cinnamic acid derivatives with anti-amyloid activity: champagne wine extracts contain many compounds, including ferulic acid and other cinnamic acid derivatives. The amount of total hydroxycinnamates is about 60 mg/L in red wines and 130 mg/L in white wines [43]. Champagne wine extract rich in these compounds has antioxidant activity [44] and neuroprotective effects [45].

## 5. Conclusions

Possible applications of the obtained results include the use of coffee drinks and drugs derived from them for the prevention and treatment of amyloid diseases. At the present moment, the limitation of this study is that it should be followed by testing the effect of coffee on large groups of people for the prevention of neurodegenerative diseases, and specific studies are required for valid recommendations. Moreover, ferulic acid exists as an approved food supplement and several studies confirm its neuroprotective effect in in vivo models of Alzheimer’s and Parkinson’s diseases [46,47,48], suggesting anti-amyloid, antioxidant, and anti-inflammatory effects.

Taking into account the known data on the increase of 3,4-dimethoxycinnamic acid concentration in the blood of people after coffee consumption, one can make an assumption that coffee-based beverages can be a means of preventing synucleinopathies such as Parkinson’s disease.

## Figures and Tables

**Figure 1 biomedicines-10-02255-f001:**
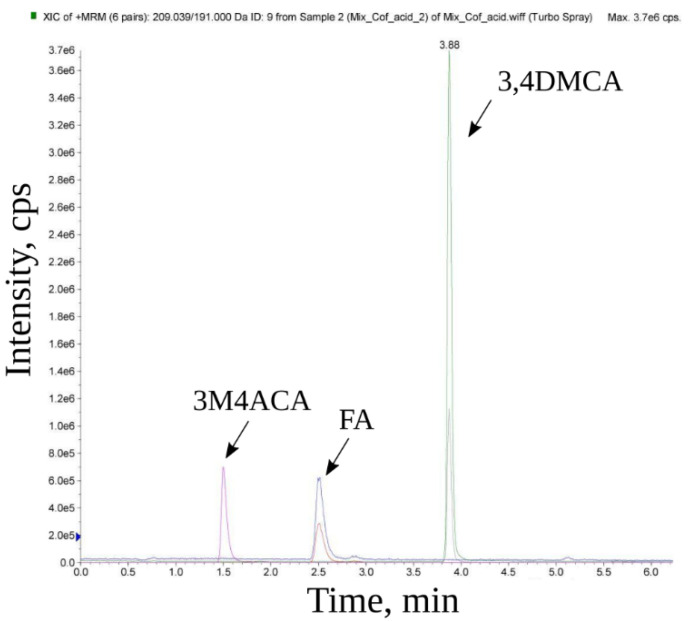
Chromatograms of the MRM transitions corresponding to the three cinnamic acid derivatives: 3-methoxy-4-acetamidoxycinnamic (3M4ACA), ferulic (FA), and 3,4-dimethoxycinnamic (3,4DMCA) acids. Two MRM transitions for each compound are shown in the chromatograms. The area of the chromatographic peak is proportional to the concentration of the substance; for the quantitative analysis a higher intensity MRM transition was used, and a second MRM transition was used for reliable identification of the compound.

**Figure 2 biomedicines-10-02255-f002:**
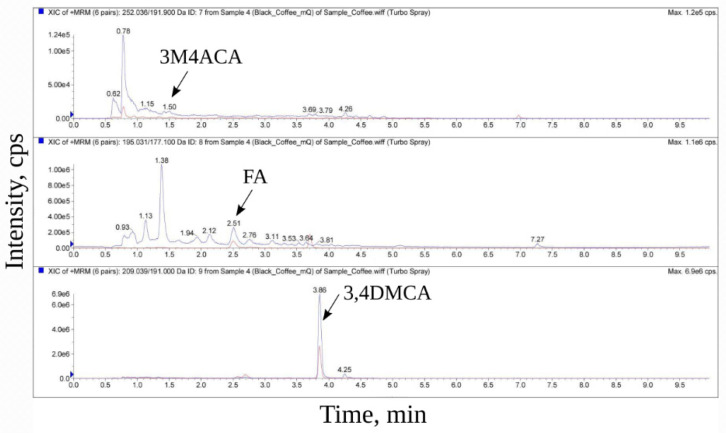
Chromatograms of aqueous extract of black coffee. Only one spectrum of aqueous extract of black coffee is shown in the figure. The spectra of the samples of other extracts are given in the Appendix A. Two MRM transitions (qualifier and quantification transition peak) are shown for each compound. The horizontal axis shows the retention time (RT) of the compounds starting from 0 to 9.5 min with 0.5 min step, and the vertical axis shows the signal intensity. In order to focus separately on the most and least intense “peaks” and “peaks” of average intensity, three different scales of these “peaks” of the same spectrum are shown in the figure; the scaling range is reflected on the vertical axis.

**Figure 3 biomedicines-10-02255-f003:**
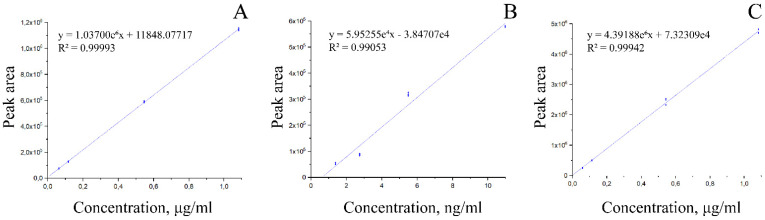
Calibration curves of hydroxycinnamic acid derivatives obtained by liquid chromatography. The vertical axis corresponds to the chromatographic signal (peak area) and the horizontal axis shows the concentrations of the calibration solutions of 3-methoxy-4-acetamidocinnamic acid (**A**), ferulic acid (**B**), and 3,4-dimethoxycinnamic acid (**C**).

**Figure 4 biomedicines-10-02255-f004:**
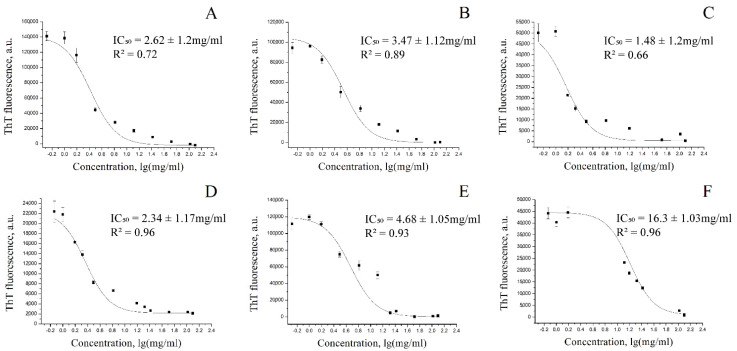
Inhibition of alpha-synuclein fibrillation by coffee extracts using thioflavin T fluorescence assay. ThT fluorescence intensity of aqueous extract of black coffee (**A**); aqueous extract of black coffee obtained by boiling (**B**); ethanol extract of black coffee (**C**); aqueous extract of green coffee (**D**); aqueous extract of green coffee obtained by boiling (**E**); and ethanol extract of green coffee (**F**).

**Table 1 biomedicines-10-02255-t001:** Conditions for MRM transitions in the positive ion mode. Two MRM transitions are used for each substance for quantitative analysis and qualitative confirmation. Technical parameters for working with samples: DP—declustering potential, EP—entrance potential, CE—collision energy, CXP—collision cell exit potential.

Substance	MRM Transition		DP (Volts)	EP (Volts)	CE (Volts)	CXP (Volts)
Ferulic acid	195.031	177.1	1	10	15	14
Ferulic acid	195.031	89	1	10	39	14
3,4-Dimethoxycinnamic acid	209.039	191	1	10	15	14
3,4-Dimethoxycinnamic acid	209.039	163	1	10	27	20
3-Methoxy-4-acetamidoxycinnamic acid	252.036	191.9	1	10	29	16
3-Methoxy-4-acetamidoxycinnamic acid	252.036	207	1	10	15	18

**Table 2 biomedicines-10-02255-t002:** 3,4DMCA derivatives and curcumin.

1	3-Methoxy-4-acetamidoxycinnamic acid	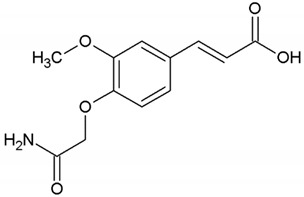
2	Ferulic acid (3-methoxy-4-hydroxycinnamic acid)	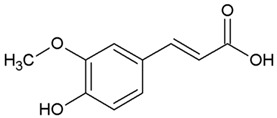
3	Dimethoxycinnamic acid (3,4DMCA)	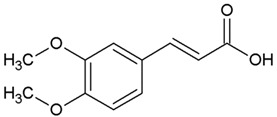
4	Curcumin	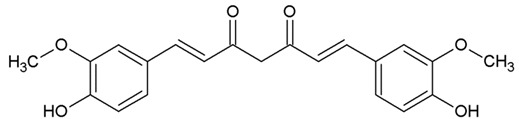

**Table 3 biomedicines-10-02255-t003:** Quantitative content of 3 cinnamic acid derivatives in coffee extracts: 3M4ACA—3-methoxy-4-acetamidoxycinnamic acid; FA—ferulic acid; and 3,4DMCA—3,4-dimethoxycinnamic acid.

Sample Name	3,4DMCA, µg/mL	3M4ACA, µg/mL	FA, ng/mL
Black coffee, mQ water	53.26	0.81	0.22
Black coffee, mQ water, heating	48.51	0.75	0.21
Black coffee, ethanol	44.06	0.40	0.13
Green coffee, mQ water	25.18	0.02	1.29
Green coffee, mQ water, heating	25.92	0.05	3.02
Green coffee, ethanol	12.96	0.01	1.55

**Table 4 biomedicines-10-02255-t004:** Values of the alpha-synuclein fibrillization inhibition constant for 3,4DMCA in different coffee extracts.

Sample Name	Ki for 3,4DMCA, μM
Black coffee, mQ water	1.12 ± 0.1
Black coffee, mQ water, heating	1.81 ± 0.17
Black coffee, ethanol	0.89 ± 0.04
Green coffee, mQ water	1.81 ± 0.09
Green coffee, mQ water, heating	4.47 ± 0.6
Green coffee, ethanol	2.88 ± 2.64

## Data Availability

Not applicable.

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
