# Peer review of "Hydroxycinnamic Acid Derivatives from Coffee Extracts Prevent Amyloid Transformation of Alpha-Synuclein"

_biomedicines, 2022, doi:10.3390/biomedicines10092255_

Round 1
Reviewer 1 Report
This paper describes effect of coffee extracts on amyloid transformation of alpha-synuclein. Unfortunately, purpose of this work is not very clear, because 3-methoxy-4-hydroxycinnamic acid and 3,4-dimethoxycinnamic acid noted here are contained in coffee and are reported to exhibit the activity (reference 11). Authors might have attempted to know optimal processing of coffee for better activity. Alternatively, authors might have tried to find other active substances from coffee. Without such clear discussions and evaluation of results, this manuscript is premature. This referee cannot recommend publication of this paper in Biomedicines.
Author Response
Dear Reviewer,
We are very grateful to you for the attention and thorough analysis of our work. We tried to answer the asked questions in detail. We hereby present our answers to the asked questions. Please find the attached initial and edited versions of the article.
Comments and Suggestions for Authors
This paper describes effect of coffee extracts on amyloid transformation of alpha-synuclein. Unfortunately, purpose of this work is not very clear, because 3-methoxy-4-hydroxycinnamic acid and 3,4-dimethoxycinnamic acid noted here are contained in coffee and are reported to exhibit the activity (reference 11). Authors might have attempted to know optimal processing of coffee for better activity. Alternatively, authors might have tried to find other active substances from coffee. Without such clear discussions and evaluation of results, this manuscript is premature. This referee cannot recommend publication of this paper in Biomedicines.
We regret that the goals and objectives of the work were not clearly formulated in the article. We have changed the abstract of the article and made additions to the text of the article. As correctly noted by the reviewer, it was previously shown that 4-hydroxy-3-methoxycinnamic acid (ferulic acid) and 3,4-dimethoxycinnamic as well as 3-methoxy-4-acetamidoacetic acid have the ability to prevent the amyloid transformation of alpha-synuclein. This was demonstrated in our previous work (Medvedeva, M.; Barinova, K.; Melnikova, A.; Semenyuk, P.; Kolmogorov, V.; Gorelkin, P.; Erofeev, A.; Muronetz, V. Naturally Occurring Cinnamic Acid Derivatives Prevent Amyloid Transformation of Alpha-Synuclein. Biochimie 2020, 170, 128–139, doi:10.1016/j.biochi.2020.01.004.). Some cinnamic acid derivatives have a similar effect on prion protein amyloidization, which was also shown by us (Zanyatkin I, Stroylova Y, Tishina S, Stroylov V, Melnikova A, Haertle T, Muronetz V. Inhibition of Prion Propagation by 3,4-Dimethoxycinnamic Acid. Phytother Res. 2017 Jul;31(7):1046-1055. doi: 10.1002/ptr.5824). The aim of this work was to find out whether these three compounds are present in coffee extracts and whether they prevent amyloid transformation of alpha-synuclein not in its individual form but as part of a complex mixture. In addition, we wanted to show that extracts obtained by different methods - ethanol and aqueous extraction, with or without heating, with treatment of green and roasted beans - have such anti-amyloid action. That is, we did exactly what the reviewer suggests (“Authors might have attempted to know optimal processing of coffee for better activity”). These experiments showed that the three studied compounds, because of their high water solubility, can be extracted even in water without heating from both types of grains. This makes it possible to propose optimal ways of obtaining coffee extracts containing cinnamic acid derivatives. We would also like to note that we were able to find in coffee extracts 3-methoxy-4-acetamidoxycinnamic acid. We used synthetic 3-methoxy-4-acetamidoxycinnamic acid in our previous works, and the fact that this compound is present in coffee extracts was not known before.
Reviewer 2 Report
The manuscript entitled “ Hydroxycinnamic acid derivatives from coffee extracts prevent 2 amyloid transformation of alpha-synuclein” is well written and organized. But the major concern is referring to the availability for some closely published articles in this regard like “ Naturally occurring cinnamic acid derivatives prevent amyloid transformation of alpha-synuclein” , the study needs to inform readers about the main aim and novelty and the reasons to apply the same points which are previously demonstrated in different researches.
- In the abstract
- In introduction : “ In our works, using the methods of circular dichroism, molecular docking, 55 thioflavin T fluorescence, proteinase K proteolysis, and ion-conducting microscopy, we 56 also showed that 3 of 9 studied synthetic and natural hydroxycinnamic acid derivatives 57 effectively prevented alpha-synuclein amyloid transformation” if this your previous study , you should mention this in the discussion and tell us the differences and relation between these studies.
- The aim of the work needs more clarification , I found it so short.
- What is the relation between coffee extracts in this study and Champagne wine extracts
- What is the sufficient amount of cinnamic acid to give the body all the benefits you mentioned and from how many cups of coffee was obtained , what type of coffee “green or roasted”
- Is there any available drugs or food supplements for these acids that are used as anti-amyloid agents suitable for the prevention and treatment of neurodegenerative amyloidosis.
- The related references for Preparation of coffee extracts were missed.
- half-maximum inhibitory concentration (IC50) is an important part of the assessment, please write it in the abstract and discuss the results in details in the discussion. The citation in the method is missed.
- All the obtained results should be discussed with the interpretations and limitations of the study.
- Recommendations and possible applications also should be added.
Author Response
Dear Reviewer,
We are very grateful to you for the attention and thorough analysis of our work. We tried to answer the asked questions in detail. We hereby present our answers to the asked questions. Please find the attached initial and edited versions of the article.
Comments and Suggestions for Authors
The manuscript entitled “Hydroxycinnamic acid derivatives from coffee extracts prevent 2 amyloid transformation of alpha-synuclein” is well written and organized. But the major concern is referring to the availability for some closely published articles in this regard like “Naturally occurring cinnamic acid derivatives prevent amyloid transformation of alpha-synuclein”, the study needs to inform readers about the main aim and novelty and the reasons to apply the same points which are previously demonstrated in different researches.
- In the abstract-
The reviewer had no comments on the abstract, but we changed it according to another reviewer's comments.
- In introduction: “In our works, using the methods of circular dichroism, molecular docking, thioflavin T fluorescence, proteinase K proteolysis, and ion-conducting microscopy, we also showed that 3 of 9 studied synthetic and natural hydroxycinnamic acid derivatives effectively prevented alpha-synuclein amyloid transformation” if this your previous study , you should mention this in the discussion and tell us the differences and relation between these studies.
The required link and discussion have been added.
- The aim of the work needs more clarification, I found it so short.
We rewrote and clarified the aim of the work and abstract.
- What is the relation between coffee extracts in this study and Champagne wine extracts
Indeed, Champagne wine extracts contain many phenolic compounds, including ferulic acid and other cinnamic acid derivatives, which may also have anti-amyloid effects. Information on the possibility of using Champagne wine extracts as another source of cinnamic acid derivatives with anti-amyloid effects has been mentioned in the Introduction, we extended it in the text of the article.
- What is the sufficient amount of cinnamic acid to give the body all the benefits you mentioned and from how many cups of coffee was obtained, what type of coffee “green or roasted”
Our results indicate that both green and roasted coffee extracts have comparable anti-amyloid activity, but it is very difficult to adequately estimate a sufficiently effective number of cups of coffee. In the initial stages of neurodegenerative diseases amyloid aggregation is a random and very long process (years!), so regular and long-term consumption of safe food products like coffee is promising to prevent neuronal damage and death. Full-fledged studies are needed to find therapeutic doses of cinnamic acids.
- Is there any available drugs or food supplements for these acids that are used as anti-amyloid agents suitable for the prevention and treatment of neurodegenerative amyloidosis.
Ferulic acid exists as approved food supplement and there are several studies confirming its neuroprotective effect in vivo models of Alzheimer and Parkinson diseases (Ojha, S.; Javed, H.; Azimullah, S.; Abul Khair, S.B.; Haque, M.E. Neuroprotective Potential of Ferulic Acid in the Rotenone Model of Parkinson’s Disease. Drug Des Devel Ther 2015, 9, 5499–5510, doi:10.2147/DDDT.S90616; Wang, E.-J.; Wu, M.-Y.; Lu, J.-H. Ferulic Acid in Animal Models of Alzheimer’s Disease: A Systematic Review of Preclinical Studies. Cells 2021, 10, 2653, doi:10.3390/cells10102653; Sgarbossa, A.; Giacomazza, D.; di Carlo, M. Ferulic Acid: A Hope for Alzheimer’s Disease Therapy from Plants. Nutrients 2015, 7, 5764–5782, doi:10.3390/nu7075246.), suggesting anti-amyloid, antioxidant and anti-inflammatory effect. We added this information to the Discussion.
- The related references for Preparation of coffee extracts were missed.
We used common recipes for preparation of coffee extracts, like “turkish” coffee with cezve, and classical recipes for ethanol extraction from plants.
- half-maximum inhibitory concentration (IC50) is an important part of the assessment, please write it in the abstract and discuss the results in details in the discussion. The citation in the method is missed.
A detailed analysis of the data obtained to determine the half-maximum inhibitory concentration (IC50) has been added to the discussion, the concentrations themselves are listed in the abstract. The missing reference has been added.
- All the obtained results should be discussed with the interpretations and limitations of the study.
A discussion of the results has been added to the text of the article.
- Recommendations and possible applications also should be added.
We added recommendations and possible applications.
Round 2
Reviewer 1 Report
Purpose of this work is well described. This referee has several scientific questions. For what experiment synthetic compounds were used, authentic samples? Although IC50 of these cinnamic acids are in the order of 10-100 microM as noted in introduction, data obtained in this study are in the order of 1 microM (Table 4). Such note and preferably more discussion are needed on this phenomenon (line 386-389). This manuscript can be published in Biomedicines with suitable revisions. Additional comments follow.
1) Introduction, line 22. What “synthetic 3- methoxy-4-acetamidoacetic acid (0.4-0.8 μg/ml) was detected in the roasted coffee extracts” means?
2) Line 258?
3) Figure 2. Horizontal axis, numbers cannot see.
Author Response
Dear Reviewer,
We are very grateful to you for the attention and further analysis of our work. We tried to answer the asked questions in detail. We hereby present our answers to the asked questions. Please find the attached initial and edited versions of the article.
Reviewer 1
Purpose of this work is well described. This referee has several scientific questions. For what experiment synthetic compounds were used, authentic samples? Although IC50 of these cinnamic acids are in the order of 10-100 microM as noted in introduction, data obtained in this study are in the order of 1 microM (Table 4). Such note and preferably more discussion are needed on this phenomenon (line 386-389). This manuscript can be published in Biomedicines with suitable revisions. Additional comments follow.
1) Introduction, line 22. What “synthetic 3- methoxy-4-acetamidoacetic acid (0.4-0.8 μg/ml) was detected in the roasted coffee extracts” means?
2) Line 258?
3) Figure 2. Horizontal axis, numbers cannot see.
For what experiment synthetic compounds were used, authentic samples?
We used two samples of natural derivatives of cinnamic acid, 3,4DMCA and ferulic acid, as standards in our mass spectrometric experiments. These samples were obtained by our colleagues by synthesis. Their identity to natural compounds was proved by them earlier. In addition, the synthetic compound 3- methoxy-4-acetamidoxycinnamic acid was used.
Although IC50 of these cinnamic acids are in the order of 10-100 microM as noted in introduction, data obtained in this study are in the order of 1 microM (Table 4). Such note and preferably more discussion are needed on this phenomenon (line 386-389).
We have added a possible explanation for this effect. According to Table 4, the IC50 for 3,4DMCA is between 0.9 and 4.5 microM. Which, although close to the values obtained for the pure compound, is still significantly lower. Probably, other cinnamic acid derivatives and various compounds present in coffee extracts have an additional effect. This is indicated by different IC50 for different types of extracts (Table 4). It is likely that the composition of compounds exerting anti-amyloid action in these extracts is different. Reasoning on this point is added to the text.
1) Introduction, line 22. What “synthetic 3- methoxy-4-acetamidocinnamic acid (0.4-0.8 μg/ml) was detected in the roasted coffee extracts” means?
This phrase has been changed and clarified. It was meant that a compound identical to 3- methoxy-4-acetamidoxycinnamic acid was found in the extracts, which was obtained by synthesis and had not previously been found in natural sources.
2) Line 258?
We fixed the letters missing because of a layout defect when adding the Figure 2.
3) Figure 2. Horizontal axis, numbers cannot see.
The retention time scale has been added to the figure description because it is technically complicated to change the font size on the spectra.
Reviewer 2 Report
- The manuscript needs substantial language revision. - What is the aim of Fibrillation of alpha-synuclein , what is the references for this method , Discuss this point throughly in the discussion section. - Please add a section contains all recent researches or most about accumulated evidences on confirming the neuroprotective effects of coffee and some of its components and relation to the amyloide. - The limitations of the study should be stated.
Author Response
Dear Reviewer,
We are very grateful to you for the attention and further analysis of our work. We tried to answer the asked questions in detail. We hereby present our answers to the asked questions. Please find the attached initial and edited versions of the article.
Reviewer 2
Comments and Suggestions for Authors
- The manuscript needs substantial language revision. - 1. What is the aim of Fibrillation of alpha-synuclein, what is the references for this method, Discuss this point throughly in the discussion section. - 2. Please add a section contains all recent researches or most about accumulated evidences on confirming the neuroprotective effects of coffee and some of its components and relation to the amyloide. - 3. The limitations of the study should be stated.
The manuscript needs substantial language revision
The text was edited to take into account the recommendations of the native speaker.
- What is the aim of Fibrillation of alpha-synuclein, what is the references for this method, Discuss this point throughly in the discussion section.
Synucleinopathies are characterized by the accumulation of alpha-synuclein. Fibrillation of alpha-synuclein in vitro allows us to partially simulate these conditions and test the effect of coffee extracts on the formation of fibrils. There are various protocols for obtaining alpha-synuclein fibrils, the method used in this article, we used earlier in our works [Barinova, K.V.; Kuravsky, M.L.; Arutyunyan, A.M.; Serebryakova, M.V.; Schmalhausen, E.V.; Muronetz, V.I. Dimerization of Tyr136Cys Alpha-Synuclein Prevents Amyloid Transformation of Wild Type Alpha-Synuclein. Int J Biol Macromol 2017, 96, 35–43, doi:10.1016/j.ijbiomac.2016.12.011.]. References have been added to the Materials and Methods section “Fibrillation of alpha-synuclein”.
- Please, add a section contains all recent researches or most about accumulated evidences on confirming the neuroprotective effects of coffee and some of its components and relation to the amyloide.
There are so many papers devoted to the neuroprotective effects of coffee and some of its components and relation to the amyloid. We have added several papers in response to reviewers' comments during the first revision of the article. We plan to do a fuller review of these papers in our forthcoming review article on this topic. In this experimental paper, we have added references to the recent studies related to amyloide.
- The limitations of the study should be stated.
We have added to the text of the paper our ideas about the limitations of this and similar work in the first revision of the work and we have changed this section slightly in this revision. Our work is the first stage, which in principle should be followed by the stage of testing the effect of coffee on large groups of people on the prevention of neurodegenerative diseases, Unfortunately, this kind of research is beyond our competence. But we hope that the data obtained in this and previous works on the effect of coffee extracts and their components will be useful in such works.